# Measuring the Invisible: Microbial Diagnostics for Periodontitis—A Narrative Review

**DOI:** 10.3390/ijms262010172

**Published:** 2025-10-19

**Authors:** Michihiko Usui, Suzuka Miyagi, Rieko Yamanaka, Yuichiro Oka, Kaoru Kobayashi, Tsuyoshi Sato, Kotaro Sano, Satoru Onizuka, Maki Inoue, Wataru Fujii, Masanori Iwasaki, Wataru Ariyoshi, Keisuke Nakashima, Tatsuji Nishihara

**Affiliations:** 1Division of Periodontology, Kyushu Dental University, Kitakyushu 803-8580, Japan; r25miyagi@fa.kyu-dent.ac.jp (S.M.);; 2School of Oral Health Sciences, Kyushu Dental University, Kitakyushu 803-8580, Japan; 3Division of Infections and Molecular Biology, Kyushu Dental University, Kitakyushu 803-8580, Japanarikichi@fa.kyu-dent.ac.jp (W.A.); 4Division of Dysphagia Rehabilitation, Kyushu Dental University, Kitakyushu 803-8580, Japan; 5Division of Preventive Dentistry, Department of Oral Health Science, Graduate School of Dental Medicine, Hokkaido University, Sapporo 060-8586, Japan; iwasaki@den.hokudai.ac.jp; 6Faculty of Dentistry, Kyushu Dental University, Kitakyushu 803-8580, Japan

**Keywords:** periodontitis, microbial diagnostics, enzymatic activity assay, real-time PCR, immunochromatography, ELISA, microarray, metagenomic analysis

## Abstract

Periodontitis is a biofilm-driven inflammatory disease in which conventional indices (probing depth, clinical attachment level, and radiographs) quantify tissue destruction without capturing the biology of infection. In this review, we synthesized microbiological diagnostics, from chairside tools to omics. We outline sampling strategies and emphasize the quantitative monitoring of bacterial load. Enzymatic assays (e.g., N-benzoyl-DL-arginine-2-naphthylamide hydrolysis assay test) measure functional activity at the point of care. Immunological methods include rapid immunochromatography for *Porphyromonas gingivalis* and enzyme-linked immunosorbent assay for the high-throughput measurement of bacterial antigens. Molecular platforms encompass quantitative polymerase chain reaction (qPCR) (TaqMan, SYBR, multiplex panels; propidium monoazide quantitative-qPCR for viable cells), checkerboard DNA–DNA hybridization for semi-quantitative community profiling, loop-mediated isothermal amplification (LAMP)/molecular beacon-LAMP for portable isothermal detection, and microarrays. Complementary modalities such as fluorescent in situ hybridization, next-generation sequencing, and Fourier transform infrared spectroscopy provide spatial, ecological, and biochemical resolutions. We discuss the limitations of current approaches, including sampling bias, presence–activity discordance, semi-quantitation, method biases, limited strain/function resolution, low-biomass artifacts, and lack of validated cutoffs. To address these challenges, we propose a pragmatic hybrid strategy: site-specific quantitative panels combined with activity and host-response markers interpreted alongside clinical metrics under standardized quality assurance/quality control. Priorities include outcome-linked thresholds, strain-aware/functional panels, robust point-of-care chemistry, and harmonized protocols to enable personalized periodontal care.

## 1. Introduction

Periodontal disease is a chronic inflammatory condition caused by periodontopathic bacteria, leading to the destruction of periodontal tissues such as the cementum, periodontal ligament, and alveolar bone. As this destruction progresses, teeth become mobile and may eventually be lost [1]. Tooth loss due to periodontitis has been reported to reduce masticatory function and impair the quality of life [2].

Recently, advances in “periodontal medicine” have revealed that periodontitis is not merely a localized oral inflammatory disease; however, it is associated with the pathogenesis and progression of various systemic conditions [3,4,5]. Periodontitis has been linked to varying degrees with systemic diseases, such as diabetes mellitus [6,7], chronic kidney disease [8,9], and adverse pregnancy outcomes, including preterm birth and low birth weight [10,11]. It is thought that bacteria and their toxins accumulated in periodontal tissues, as well as inflammatory mediators produced locally, may disseminate systemically and influence these conditions [12].

Clinically, periodontitis is diagnosed by the presence of attachment loss, which develops through the following sequence: Connective tissue attachment loss, a detachment of collagen fibers anchoring the gingiva to the tooth root from the cementum; Junctional epithelium migration, an apical displacement of the junctional epithelium, which normally forms a protective seal at the gingival margin; and consequently, Periodontal pockets, which are deepened spaces between the tooth and gingiva that are prone to bacterial colonization [13,14]. Anaerobic periodontopathic bacteria proliferates within the periodontal pockets, triggering inflammatory alveolar bone resorption [1].

Thus, clinical examination for periodontitis typically includes patient interview, measurement of probing pocket depth (PPD), tooth mobility, and radiographic assessment of alveolar bone levels. However, these conventional diagnostic parameters only provide an evaluation of the extent of structural destruction and offer no direct information regarding the presence, type, or activity of the causative periodontopathic bacteria.

The human oral cavity harbors several hundred bacterial species [15]. These microorganisms form biofilms and colonize tooth surfaces and periodontal pockets. Among them, periodontopathic bacteria collectively refer to the bacterial species involved in the onset and progression of periodontal diseases. Of these, three species—*Porphyromonas gingivalis* (*P. gingivalis*), *Tannerella forsythia* (*T. forsythia*), and *Treponema denticola* (*T. denticola*)—comprising the so-called “red complex” possess particularly high pathogenic potential [16].

Periodontopathic bacteria exert their virulence through a variety of mechanisms, including cytotoxic factors such as lipopolysaccharide, leukotoxin, cytolethal distending toxin, trypsin-like proteases, dentilisin, and adhesion factors such as fimbriae, bacteroides surface protein A (BspA), major surface protein, and fusobacterium adhesion A [17,18,19,20,21]. The proliferation of these organisms can disrupt the composition and balance of the oral microbiota, an ecological shift known as dysbiosis, which plays a critical role in the initiation of periodontal disease [22,23].

In the context of periodontal dysbiosis, *P. gingivalis* is a keystone pathogen that can subvert the host immune system and impair leukocyte function [24,25,26]. The primary and most essential approach in periodontal therapy is the removal and control of bacterial plaques, including periodontopathic species, to halt disease progression. Nevertheless, PPD is routinely measured as part of the treatment outcome assessment; however, direct microbiological examination for periodontopathic bacteria is still rarely performed in clinical practice. Such examinations have the potential to detect “sites with high disease activity” and “individuals at increased risk of disease onset” that may not be identified through clinical parameters alone, such as PPD. Furthermore, they can be utilized to inform the indication and subsequent evaluation of antimicrobial therapy or local drug delivery. Incorporating microbiological testing into periodontal therapy may therefore enable a more refined and evidence-based approach. In this review, we focus on the potential role of microbiological testing for periodontopathic bacteria.

This review provides an overview of the contemporary assays for the detection of periodontopathic bacteria. The surveyed methods span a wide spectrum, from commercially available chairside tests to laboratory-based platforms. In addition, we critically appraise the principal limitations and methodological challenges of these approaches and discuss emerging technologies and future directions for their clinical translation.

## 2. Detection Methods for Periodontopathic Bacteria

Various methods for detecting periodontopathic bacteria have been reported, and representative approaches are summarized in Table 1. As noted above, periodontitis results from a complex interplay between multiple periodontopathic species and disruption of the normal oral microbiota. However, because periodontopathic bacteria are often present, even in healthy gingival sulci, purely qualitative testing, which simply confirms the presence or absence of a target species, has limited clinical value. Therefore, in the context of periodontal disease, quantitative assays that measure the bacterial load are of great importance. Monitoring changes in the number of periodontopathic bacteria before and after treatment is a fundamental approach for evaluating therapeutic outcomes.

### 2.1. Sample Collection Techniques

The type of specimen used for the detection of periodontopathic bacteria varies according to the purpose of the examination. For example, saliva or tongue-coating samples are appropriate when assessing the carriage level of a specific periodontopathic species within the oral cavity. In contrast, when analyzing local periodontal status, sampling should be performed from the subgingival plaque or the gingival crevicular fluid at the site of interest.

For the purpose of microbiological analysis, specimen collection 5 carried out as follows. For tongue coating samples, the entire dorsal surface of the tongue was swabbed using a sterile, dedicated cotton swab, moving from the posterior to the anterior region approximately ten times. For subgingival plaque sampling, rolled cotton was placed around the target tooth to isolate it from the saliva. Supragingival plaque was carefully removed using an excavator. A sterile #40 paper point was inserted into the designated periodontal pocket, left in place for 30 s, and then removed. The retrieved paper points were subsequently used as test specimens.

### 2.2. Enzymatic Activity Assays

Methods that directly measure the enzymatic activity of periodontopathic bacteria are of particular importance because they allow evaluation of the functional activity of bacterial virulence. A representative example is the N-benzoyl-DL-arginine-2-naphthylamide hydrolysis assay (BANA) (BANAMet LLC, Ann Arbor, MI, USA), which detects trypsin-like protease activity produced by the Red Complex species *P. gingivalis*, *T. denticola*, and *T. forsythia*. In this assay, bacterial proteases hydrolyze the synthetic substrate to release 2-naphthylamide [27,28,29,30]. A positive reaction has been reported to correlate with PPD and clinical attachment loss. This assay has been applied clinically as a simple and rapid chairside diagnostic method [31].

ADCHECK^®^ (ADTEC Co., Ltd., USA, Japan), a diagnostic kit for detecting trypsin-like enzyme activity produced by Red Complex species, has been developed and applied in clinical and epidemiological studies. ADCHECK^®^ does not require an incubator and can be performed at room temperature, unlike the BANA test. It can detect as few as 1000 *P. gingivalis* cells and has a high concordance rate compared with real-time PCR (Figure 1) [32]. The kit assigns a score to the intensity of trypsin-like enzyme activity, which has been reported to correlate with periodontal disease status and is useful for screening severe periodontitis [33,34,35].

Recently, a simple device capable of real-time detection of trypsin-like peptidase activity within individual periodontal pockets has been developed, offering promise for site-specific assessment of disease status and monitoring of treatment outcomes [36]. Furthermore, studies have been conducted to measure protease activity specific to certain bacterial species, such as dentilisin from *T. denticola*, using fluorogenic or chromogenic substrates [37]. These enzymes directly contribute to extracellular matrix degradation and immune evasion, making them valuable indicators of bacterial activity.

Commercial kits for measuring the enzymatic activity of periodontopathic bacteria are valued by clinicians for being faster than culture or PCR and are particularly effective for patient education and treatment motivation [38,39,40]. However, because their performance depends on factors such as detection sensitivity, specificity, and operating conditions (e.g., temperature control), the results should be interpreted in conjunction with clinical findings and other diagnostic tests [40].

The relationships of enzymatic activity with probing depth and periodontitis diagnosis are shown in Table 2.

### 2.3. Immunological Methods

#### 2.3.1. Immunochromatography

Immunochromatography has gained attention as a rapid and simple method for detecting *P. gingivalis*, the principal pathogen in periodontitis. This immunoassay employs antibodies within a test cassette to allow the visual detection of the presence or absence of the target antigen. Imamura et al. developed an immunochromatography device for detecting *P. gingivalis* in subgingival plaque samples and demonstrated that the test results (band scores) significantly correlated with the PPD and clinical attachment level (CAL), indicating its utility in evaluating treatment outcomes [41]. O’Brien-Simpson et al. reported that immunochromatography has a high diagnostic performance as a chairside rapid test for assessing the activity of *P. gingivalis* [42]. Yamanaka et al., reported that an immunochromatographic device can be used to detect *P. gingivalis* in subgingival plaques, showing a significant correlation with real-time PCR results (correlation coefficient = 0.73, *p* < 0.005) together with high specificity (98%), positive predictive value (94%), negative predictive value (89%), and overall accuracy (90%) (Figure 2) [43].

Table 3 summarizes studies evaluating immunochromatographic assays for *Porphyromonas gingivalis* in relation to probing pocket depth and periodontal disease status.

Owing to advantages such as low cost, speed, and non-invasiveness, immunochromatography is considered a promising approach for periodontal disease screening. Future developments are expected to focus on multiplex biomarker detection and improved quantification, further enhancing diagnostic accuracy and clinical applicability [44,45].

#### 2.3.2. Enzyme-Linked Immunosorbent Assay

Enzyme-Linked Immunosorbent Assay (ELISA) is an immunological method that enables sensitive and specific detection of antigens derived from periodontopathic bacteria or host antibodies. The antigen or antibody titer can be determined by measuring the intensity of the colorimetric reaction, using immobilized antigens (or antibodies) and enzyme-labeled antibodies.

*Aggregatibacter actinomycetemcomitans* (*A. actinomycetemcomitans*) is the major pathogen involved in aggressive periodontitis. Gadekar et al. compared serum and salivary immunoglobulin (Ig)G and IgA levels between patients with chronic periodontitis and healthy controls using indirect ELISA and reported significantly higher values in the patient group [46]. Lakio et al. monitored serum and salivary IgG antibody levels against *A. actinomycetemcomitans* and *P. gingivalis* for over 15 years and revealed that antibody titers remained relatively stable within individuals [47]. For *T. forsythia*, antibody levels against the virulence factor BspA have been associated with clinical parameters, and Hall et al. found that ELISA-measured anti-BspA antibody levels correlated significantly with clinical attachment loss [48]. Furthermore, Kakuta et al. evaluated antibody titers against multiple species (*P. gingivalis*, *P. intermedia*, and *A. actinomycetemcomitans*) using whole-cell antigens and revealed that such assays could be applied to estimate the composition and quantity of the infecting bacterial species [49].

The advantages of ELISA include the ability to process multiple samples simultaneously, high quantitative accuracy, and compatibility with various sample types (serum, saliva, and gingival crevicular fluid). However, its limitations include potential cross-reactivity and difficulty in distinguishing past infections from current infections. Recent developments, such as chemiluminescent ELISA and multiplex platforms, have enabled highly sensitive multi-analyte simultaneous measurements, expanding the application of ELISA to periodontal disease diagnosis, epidemiological studies, and the evaluation of treatment outcomes [50,51].

### 2.4. Real-Time PCR (Quantitative PCR)

Real-time PCR (quantitative PCR [qPCR]) enables highly sensitive and specific quantification of target bacteria by monitoring fluorescence in real time and deriving threshold cycles (Ct) for absolute (standard curve) or relative (ΔΔCt) quantification. Adherence to the Minimum Information for Publication of Quantitative Real-Time PCR Experiments (MIQE) guidelines is recommended for study design and reporting to ensure inter-laboratory reproducibility by specifying the primer/probe design, reaction efficiency, control, and management of inhibitors [52].

Clinical specimens included subgingival plaque, saliva, and gingival crevicular fluid (GCF). Quantitative detection of major periodontopathogens, such as *P. gingivalis*, as well as the total bacterial load, is feasible and has been applied in diagnostic and research contexts [53,54]. Compared with culture, qPCR offers superior speed and sensitivity, facilitating the detection and enumeration of fastidious or low-abundance organisms. In subgingival plaques, qPCR correlates well with quantitative culture for *P. gingivalis* [55], and high agreement has been reported for multiple species, including *A. actinomycetemcomitans* and *T. forsythia* [56,57]. Moreover, multiplex qPCR panels allow simultaneous quantification of several species, enabling rapid site-specific profiling of the periodontal microbiota for baseline assessment and treatment monitoring [58,59]. Table 4 summarizes qPCR studies demonstrating higher bacterial loads in deeper periodontal pockets and in patients with periodontitis.

TaqMan chemistry (5′-nuclease assay) (Thermo Fisher Scientific Inc., Waltham, MA, USA) uses an internal, dual-labeled hydrolysis probe that is cleaved by the DNA polymerase 5′→3′ exonuclease activity during extension, releasing reporter fluorescence [60,61]. TaqMan is advantageous for discriminating closely related taxa owing to its high sequence specificity and suitability for multicolor multiplexing. In periodontology, TaqMan probes have been used for the quantification of *P. gingivalis* and other species, including six validated species assays that demonstrate analytical performance and clinical utility [53,59].

SYBR Green I chemistry (Applied Biosystems™, Thermo Fisher Scientific Inc., Waltham, MA, USA) relies on the fluorescence of the dye upon binding to double-stranded DNA, allowing for a low-cost, primer-only implementation. As nonspecific amplicons and primer dimers also generate signals, melt curve analysis is essential for product verification and specificity control [62,63]. Comparative studies on periodontal samples have shown that with proper primer design and melt curve validation, SYBR-based assays can achieve robust quantification alongside TaqMan-based methods [64].

A recognized limitation of DNA-targeted qPCR is the detection of DNA from non-viable cells, which may not reflect current microbial activity. Propidium monoazide (PMA)-qPCR addresses this by selectively masking DNA from membrane-compromised (dead) cells, enabling viable cell quantification. Its applicability has been demonstrated for key periodontopathogens in clinical and model systems [65,66]. Overall, qPCR provides a rapid, quantitative framework spanning baseline diagnosis, post-nonsurgical therapy follow-up, and site-specific risk assessment. Further clinical refinement is expected through multiplex panel development and continued standardization under MIQE [52,54,57,59].

### 2.5. Checkerboard DNA–DNA Hybridization

Checkerboard DNA–DNA hybridization (CKB) enables the simultaneous investigation of many clinical samples against many bacterial targets on a single membrane. In its canonical implementation, denatured DNA from each sample is immobilized in parallel lanes using a Miniblotter/MiniSlot apparatus. Subsequently, the membrane is rotated 90° and hybridized with a panel of species-specific probes in the orthogonal direction. Digoxigenin-labeled whole-genome probes with chemiluminescence are typically used for detection; however, oligonucleotide probes and alternative labels have also been described. This format offers high-throughput and semi-quantitative readouts suitable for large cohorts or multi-site studies in periodontal research [67,68].

This approach has become foundational in periodontal microbiology, allowing investigators to profile dozens of taxa across thousands of subgingival plaque specimens. In a landmark analysis of 13,261 samples, Socransky and colleagues used CKB with 40 whole-genome probes to identify reproducible “complexes” of co-colonizing species, most notably the red complex (*P. gingivalis*, *T. forsythia*, and *T denticola*), whose levels tracked strongly with pocket depth and bleeding on probing [69]. Subsequent longitudinal work applied CKB to treatment monitoring, documenting broad decreases in periodontal pathogens after scaling and root planing and related microbiological shifts to clinical responders from non-responders [70,71].

Analytically, CKB provides semiquantitative estimates of bacterial abundance by calibrating signal intensities to reference lanes spotted with defined amounts of target DNA or cultured cells. Its multiplex design (“many species × many sites”) keeps per-sample costs low and makes it practical to include reference strains and controls on every membrane, supporting robust between-run comparisons. Methodological refinements include comprehensive laboratory protocols for digoxigenin (DIG) detection and layout optimization, as well as alternative probe chemistry. For example, fluorescein-labeled whole-genome probes with chemiluminescent detection have been validated as viable substitutes for DIG, offering operational flexibility without compromising performance [68,72].

CKB has also been coupled with upstream multiple-displacement amplification (MDA) to overcome low biomass constraints and generate targets and probes from minimal input DNA. In oral biofilms and endodontic specimens, MDA-augmented CKB yielded signals and proportions comparable to those obtained from unamplified material, facilitating analyses where the sample yield was limited [73]. Beyond periodontitis, CKB has been used to quantify bacterial contamination in implant–abutment assemblies and to survey peri-implant microbiota, underscoring its broader applicability to implant dentistry [74]. The findings from checkerboard DNA–DNA hybridization studies are summarized in Table 5, showing higher prevalence of red complex species in deeper pockets and in periodontitis.

Despite its strengths, the CKB has several important limitations. Signal intensities are only semiquantitative and depend on DNA quality, hybridization conditions, and the dynamic range of detection chemistry. Whole-genome probes exhibit limited cross-hybridization, whereas oligonucleotide probes improve specificity at the expense of breadth and sometimes sensitivity. These constraints motivated careful probe validation and inclusion of internal standards and controls for each membrane [67,75]. Method comparison studies show that CKB and PCR-based assays usually agree in distinguishing health versus disease and correlate for many taxa; however, quantitative correspondence varies by species. Thus, CKB results are best interpreted alongside clinical parameters and, when needed, complementary molecular tests (e.g., qPCR or sequencing) [71].
ijms-26-10172-t005_Table 5Table 5Checkerboard DNA–DNA hybridization studies demonstrating associations with probing pocket depth and periodontal disease status.ReferenceSample SizeRelationship with PPDAssociation with Disease StatusSocransky et al., 1998(ref. [69])185 subjects, 13,261 subgingival samplesDistinct microbial complexes (red, orange, etc.) strongly associated with increasing pocket depthRed complex species linked to periodontitis; distribution patterns distinguished health vs. diseaseHaffajee et al., 1997(ref. [70])57 periodontitis patientsScaling and root planing reduced mean pocket depth; reductions correlated with decreases in red/orange complex speciesMicrobial shifts after therapy paralleled clinical improvementsdo Nascimento et al., 2009(ref. [74])30 implant patientsNot directly assessed at periodontal pockets; focus on internal implant contaminationCheckerboard detected peri-implant microbial leakage; compared cast vs. pre-machined abutmentsHaffajee et al., 2009(ref. [76])187 subjects, 9182 subgingival samplesCheckerboard and PCR showed consistent detection trends; bacterial levels increased with pocket depthBoth methods identified higher prevalence of pathogens in periodontitis compared with healthPPD, probing pocket depth.


Overall, CKB remains a workhorse for high-throughput, multisite, and multi-species profiling of subgingival communities. It is particularly valuable when the investigative goal is to map community-level patterns across the dentition or to monitor broad microbial shifts after therapy, where scalable, standardized, semiquantitative data from many species and sites are more informative than absolute quantification of a few targets [69,71,75,76].

### 2.6. Loop-Mediated Isothermal Amplification

LAMP is increasingly recognized as a powerful molecular diagnostic tool because it omits the DNA denaturation step, employs a strand-displacing DNA polymerase, achieves high target specificity through four primers, and operates under isothermal conditions that improve efficiency by eliminating thermal cycling [77,78]. Compared with conventional PCR or qPCR, LAMP offers higher analytical sensitivity, excellent specificity, operational efficiency, ease of handling, and the capability to detect multiple pathogens simultaneously. Using this approach, periodontal pathogens can be identified reliably [79,80]. Molecular isothermal loop amplification (MB-LAMP), an isothermal method that combines the advantages of LAMP and qPCR, provides greater specificity and sensitivity than conventional LAMP. Owing to its compact instrumentation, simple workflow, and rapid turnaround, LAMP is a promising platform for chairside screening and longitudinal monitoring of periodontopathogens in clinical practice [81].

### 2.7. Microarray-Based Assays

Microarrays comprise thousands of immobilized DNA probes and enable the simultaneous measurement of the relative abundance of many distinct DNA or RNA sequences within a sample by hybridization and subsequent signal detection. In periodontal microbiology, this platform has been leveraged for culture-independent identification of bacteria, broad profiling of community structures, and gene expression patterns [75,82].

The Human Oral Microbe Identification Microarray (HOMIM) (The Forsyth Institute, Cambridge, MA, USA) is a high-throughput 16S ribosomal (r)RNA–based technology developed specifically for oral microbial profiling. In a single hybridization on a glass slide, HOMIM can detect approximately 300 taxa, including uncultivable species, thus providing wide coverage of the oral ecosystem [75,82,83]. Subgingival microbial profiles have been compared across disease states using HOMIM. For example, refractory periodontitis, severe periodontitis, and periodontal health status exhibit distinct signatures in the prevalence and levels of key taxa [83]. Longitudinally, HOMIM has been used to monitor the impact of periodontal therapy, differentiating “good responders” from refractory cases by characterizing shifts in the subgingival microbiota after scaling and root planing [84]. A recent systematic review concluded that HOMIM is effective for identifying major periodontopathogens and for comparative community analyses, while noting technical considerations, such as probe design, normalization, and semi-quantitative readouts [85].

In addition to HOMIM, microarray design strategies have been proposed for complex bacterial communities, emphasizing probe selection, coverage, and specificity in mixed-species biofilms [86]. DNA microarrays tailored to oral niches, including teeth, mucosa, and dental implants, have been designed and validated for phylogenetic analyses, supporting applications spanning health, periodontitis, and peri-implant conditions [87,88]. Several clinically oriented DNA chips have been developed using focused panels of periodontopathogens. The ParoCheck^®^ DNA chip (Greiner Bio-One GmbH, Frickenhausen, Germany) is a commercial 16S rRNA–based microarray designed for clinical periodontal diagnostics that targets 10 or 20 species commonly implicated in disease [82,89]. The Oral Care Chip (Mitsubishi Chemical Corporation, Tokyo, Japan) provides targeted detection (17 species) with quantitative output and has been validated for clinical samples to support screening and monitoring in periodontal care [90].

Methodologically, microarrays offer high sample throughput and multiplex species coverage in a single assay, making them well suited for cross-sectional comparisons across many sites and subjects and for tracking broad community shifts after therapy [75,83,84,87,88,89,90]. However, as hybridization-based platforms, they are inherently semi-quantitative; dynamic range, cross-hybridization, and dependence on curated probe sets necessitate careful normalization and validation, and complementary methods (e.g., qPCR or sequencing) may be required when absolute quantification or strain-level resolution is needed [75,82,85,86,87]. Nevertheless, for panel-based surveillance of established periodontal pathogens and community-level mapping across dentition, microarrays remain a useful and pragmatic approach in both research-and clinically oriented investigations.

### 2.8. Fluorescence In Situ Hybridization

rRNA-targeted fluorescence in situ hybridization (FISH) identifies and enumerates taxa directly in clinical biofilms without cultivation while preserving the spatial context critical to periodontal pathogenesis. Methodological advances include detailed probe design, permeabilization, control, and quantitative image analysis, and enable multicolor/spectral multiplexing for complex communities [91]. In the oral cavity, combinational labeling and spectral imaging-FISH mapped micron-scale biogeography of supragingival plaque, revealing reproducible “hedgehog/corncob” consortia and niche partitioning consistent with metabolic and oxygen gradients [92]. In vivo imaging revealed the stratified architecture of natural teeth and pathogen colonization of preformed biofilms as discrete microcolonies [93]. Strengths include who-is-where resolution, and limitations include semi-quantitative signal interpretation, potential cross-hybridization, and specialized sample preparation [91,92,93].

### 2.9. Next-Generation Sequencing

16S rRNA amplicon and shotgun metagenomic profiling of the full oral microbiome, including uncultivable taxa, robustly discriminates periodontitis from healthy individuals by community-level signatures (e.g., enrichment of spirochetes, *Filifactor alocis*, red-complex species) and altered diversity [94,95]. Systematic evidence shows that next-generation sequencing (NGS) can track subgingival community shifts after therapy, enabling responder/non-responder stratification and longitudinal monitoring [96]. Recent oral-specific guidelines highlight choices that minimize bias in target regions (V3–V4/V4), read length/quality trimming, reference databases, and contamination control, thereby improving cross-study comparability and reproducibility [97]. NGS offers breadth and discovery power, and when absolute quantification or strain-level resolution is required, complementary qPCR, targeted assays, or high-depth metagenomics may be required [94,95,96,97].

### 2.10. Fourier Transform Infrared Spectroscopy

Label-free Fourier transform infrared (FTIR) “fingerprints” of saliva provide a rapid, non-invasive readout of biochemical changes linked to periodontal inflammation. FTIR microscopy with chemometrics (e.g., partial least squares-discriminatory analysis) was used to classify periodontitis from saliva in pilot clinical studies [98], whereas attenuated total reflectance-FTIR workflows revealed point-of-care feasibility and encouraging accuracy, extending to patients with comorbid diabetes [99]. Beyond host signatures, FTIR combined with machine learning can detect pathogen-specific signals (e.g., *P. gingivalis*) from oral bacterial preparations, suggesting a complementary route for direct microbial detection [100]. Larger standardized cohorts and harmonized preprocessing are required for clinical translation [98,99,100].

## 3. Discussion

In this review, we surveyed the microbiological assays for periodontitis, ranging from rapid, commercially available chairside tests to laboratory platforms that require specialized and costly instrumentation. Each modality has distinct strengths and constraints, spanning target readout (presence vs. activity), analytical sensitivity/specificity, throughput, turnaround time, cost, and interpretability; therefore, test selection should be tailored to the clinical question (screening, risk stratification, and treatment monitoring), specimen type, and local resources. In the following sections, we organize a critical appraisal of unresolved challenges, including sampling bias and site heterogeneity, semiquantitative outputs, method-specific biases and limited standardization, restricted strain/functional resolution, contamination in low-biomass specimens, and the lack of validated outcome-linked cutoffs. We also outline pragmatic solutions and future directions, such as hybrid testing strategies, rigorous quality assurance (QA)/quality control (QC) and reporting standards, and the incorporation of strain-aware and host–microbe functional metrics into clinically actionable workflows.

### 3.1. Sampling Bias and Site-Level Heterogeneity

Subgingival ecosystems are spatially patchy on a millimeter scale. The results can change meaningfully with (i) the sampling site (deepest pocket vs. bleeding site vs. random site), (ii) collection tool (paper points vs. curettes), (iii) presampling debridement of the supragingival plaque, (iv) moisture control, and (v) whether the sample was pooled across sites. Within one pocket, a 1–2 mm shift can alter the microenvironment (oxygen and nutrient flux) and, thus, the microbial profile. Saliva and tongue-coating provide attractive, non-invasive “whole-mouth” readouts but inevitably dilute site-specific signals; they are best positioned for screening or population surveillance, not definitive lesion-level decisions. Standard operating procedures (SOPs) should define isolation (e.g., cotton rolls), supragingival cleaning, insertion depth, dwell time for paper points, and replicate sampling to reduce variance and enable cross-study comparability.

### 3.2. Discordance Between Presence and Activity

Most assays, culture, PCR/qPCR, microarrays, and standard NGS, quantify nucleic acids and therefore “presence,” not real-time activity. DNA from dead cells and extracellular matrices can inflate estimates, whereas low-abundance keystone taxa can drive pathogenic shifts disproportionately to their counts. Enzyme-based tests (e.g., trypsin-like activity and dentilisin assays) and viable-cell strategies (e.g., PMA-qPCR, rRNA-targeting, and metabolomics) move closer to functional readouts; however, they also introduce caveats (substrate promiscuity and host enzyme interference, or partial loss of sensitivity). A pragmatic approach is to interpret organismal load alongside activity proxies (bacterial enzymes, short-chain fatty acids) and host response markers (e.g., matrix metalloproteinase-8, neutrophil elastase, interleukin-1β) to approximate net tissue-destructive potential.

### 3.3. Semi-Quantitation and Dynamic Range Constraints

Hybridization platforms (checkerboards and targeted arrays) are inherently semiquantitative. Cross-hybridization may blur closely related taxa, and signals depend on DNA integrity, hybridization kinetics, probe design, and detection chemistry. Chairside enzymatic tests excel in speed and patient communication; however, they are sensitive to operating conditions (temperature and incubation time), leading to center-to-center variability unless external controls and strict timing are used. Wherever feasible, include internal standards (reference lanes and synthetic spikes) and calibrate to a biologically interpretable unit (e.g., cells/site) rather than an arbitrary intensity.

### 3.4. Molecular Biases and Normalization Challenges

The 16S rRNA amplicon data were shaped by primer choice, target region (V3–V4 vs. V4), library size, and bioinformatics. Shotgun metagenomics mitigates some biases and offers functional inference; however, it demands higher input DNA, cost, and analytic expertise. qPCR delivers absolute counts if standards and reaction efficiencies are rigorously validated (per MIQE) and inhibitors are controlled; without this, cross-study synthesis is weak. Notably, amplicon data are compositional; an apparent increase in one taxon may reflect a decrease in another. Where absolute trends matter (e.g., decontamination efficacy), incorporate spike-in standards or orthogonal absolute methods (qPCR/dPCR). Batch effects (kits, lots, and sequencers) were audited and corrected statistically when possible.

### 3.5. Limited Strain-Level and Functional Resolution

Most clinical panels resolve to species, yet pathogenicity often hinges on strain-level features (e.g., fimbrial genotypes and toxin loci) and community functions (e.g., proteolysis, heme acquisition, and immunomodulation). Taxonomy alone overlooks these determinants. Metagenomics, targeted virulence PCR, or proteomic/metabolomic overlays can recover function and strain signals at an added cost. For routine care, a compromise is the use of targeted panels that integrate species and key virulence determinants with clear reporting of what is or is not captured.

### 3.6. Confounding by Host and Behavior

Smoking, diabetes, recent antibiotics/antiseptics, diet, and oral hygiene within the previous 12–24 h reshaped microbial and host response signals. The timing of sampling (diurnal variation), menstrual cycle, and recent dental procedures are also important. Studies should prespecify washout windows, record covariates, and analyze them with appropriate adjustments. For longitudinal care, samples should be obtained at consistent times and behaviors (e.g., before morning brushing/food).

### 3.7. Low-Biomass Artifacts and Contamination

Shallow pockets and GCF often yield low biomass where reagents and environmental DNA dominate. Field blanks, extraction blanks, and no-template controls are essential; without them, “rare” taxa may be contaminants. In high-throughput laboratories, dedicated clean areas, single-use disposables, and barcode-based tracking reduce cross-contamination. The reports should disclose contamination controls and computational decontamination applied.

### 3.8. Lack of Validated Clinical Cutoffs

For many targets, there are no universally accepted thresholds for predicting incident attachment loss, response to non-surgical therapy, or reactivation. ROC-based cutoff points derived cross-sectionally frequently overestimate the real-world performance. Prospective event-driven studies with calibration and external validation are required. Until then, tests should be framed as risk indicators rather than as stand-alone diagnostics.

### 3.9. Cost, Turnaround, and Workflow Fit

NGS and metagenomics provide unmatched ecological contexts; however, they entail higher costs and slower turnaround, challenging chairside decisions. Conversely, rapid enzymatic or immunochromatographic tests integrate well into appointments and support patient engagement but typically require clinical correlation or confirmatory testing to alter treatment plans. Laboratory-developed qPCR panels can strike a balance if batched efficiently and can be coupled with automated reporting.

### 3.10. Practical Implications

No single assay is sufficient across all indications. A defensible strategy is: (i) standardized, site-specific sampling; (ii) a hybrid test bundle—quantitative pathogen panel (qPCR or targeted multiplex) plus activity/host markers; (iii) contextual interpretation with clinical metrics (PPD, CAL, bleeding on probing, mobility); and (iv) rigorous QA/QC (blanks, spikes, positive/negative controls, inhibitor audits) and transparent reporting (e.g., MIQE for qPCR, detailed pipelines for NGS). Reports should avoid over-precision, prioritize actionable categories (e.g., low/moderate/high risk with confidence bounds), and flag assay limitations. From a clinical perspective, the value of microbiological testing lies in its ability to complement traditional parameters by providing biologically grounded information that can directly inform patient care. Chairside enzymatic or immunochromatographic assays may serve as motivational tools to enhance patient compliance and facilitate communication of disease activity. Quantitative molecular panels can support risk stratification, assist in determining the indication for adjunctive antimicrobial or regenerative therapy, and enable objective evaluation of treatment response. In supportive periodontal therapy, longitudinal monitoring of bacterial load and activity markers may help identify sites at risk of recurrence before clinical breakdown occurs. Accordingly, microbiological assays should be viewed not as stand-alone diagnostics but as adjunctive tools that enrich clinical decision-making and promote personalized, evidence-based periodontal care.

### 3.11. Future Directions

Priorities include: (1) prospective validation of microbiologic thresholds tied to hard outcomes (future attachment loss, tooth survival, retreatment); (2) strain-aware and function-informed panels that incorporate virulence markers and metabolic/host-response surrogates; (3) temperature-robust, controlled point-of-care chemistries with built-in external controls; (4) harmonized SOPs for sampling, storage (e.g., stabilization buffers, freeze–thaw limits), and analysis to reduce inter-center variance; and (5) integration of multimodal data (microbes + host + behavior) into calibrated prediction models that report uncertainty and avoid overfitting. Advancing along these lines will improve the interpretability and clinical utility of periodontal bacterial testing while supporting evidence-based personalized care.

## 4. Conclusions

Microbiological testing for periodontitis is most powerful when it is quantitative, functional, and standardized. No single assay suffices; combining various tests with activity and host response readouts best informs care. Validated outcome-linked thresholds and robust point-of-care strain-aware tools are the next steps toward precise risk stratification, truly personalized periodontal therapy, and cross-center comparability.

## Figures and Tables

**Figure 1 ijms-26-10172-f001:**
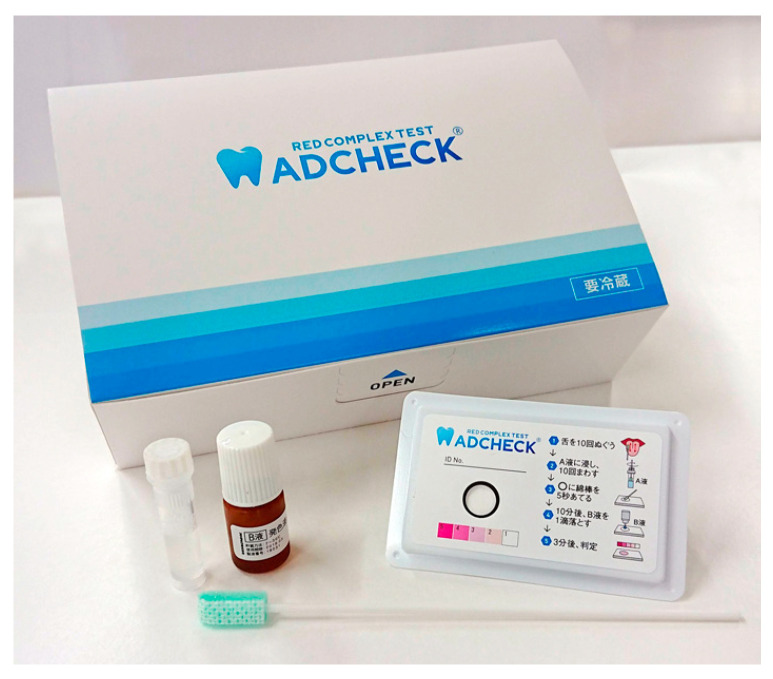
An Example of a Periodontal Pathogen Enzyme Activity Test (ADCHECK^®^) From left to right: processing solution, color-developing solution, test plate, and a cotton swab for sample collection. The test plate is small, approximately the size of a business card. The test can be performed chairside.

**Figure 2 ijms-26-10172-f002:**
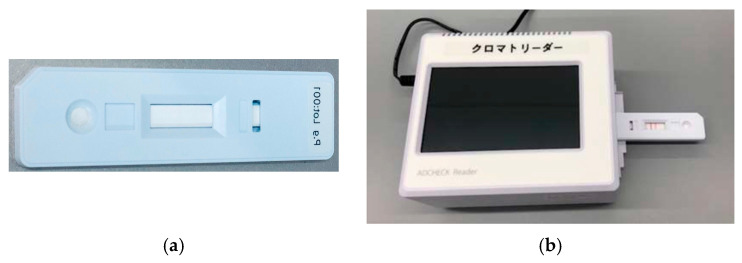
An example of an immunochromatographic test. This test kit was used in Ref. [43]. (**a**) Test cassette: compact in size, measuring approximately 2 cm by 8 cm; (**b**) Reader: enables scoring of the bands on the test cassette.

**Table 1 ijms-26-10172-t001:** Summary of common diagnostic methods for periodontal pathogens, including principles, advantages, limitations, and typical clinical applications.

Method	Principle	Advantages	Limitations	Main Clinical Applications
Culture method (aerobic/anaerobic)	Culturing specimens on selective or enriched media; identification based on colony morphology and biochemical characteristics	Well-established standard; enables antimicrobial susceptibility testing	Cannot detect fastidious or uncultivable species; labor-intensive and time-consuming	Identification of major periodontal pathogens; guidance for antibiotic selection
Enzymatic Activity Assays	Determination of enzyme activity	No specialized equipment required; cost-effective	Lower sensitivity and specificity compared with molecular methods	Identification of red complex pathogens
Immunological methods (Enzyme-Linked Immunosorbent Assay, Immunochromatography)	Detection of specific antigens using antigen–antibody reactions	Rapid turnaround; applicable to chairside testing	Possible cross-reactivity leading to false positives; semi-quantitative at best	Rapid screening for *Porphyromonas gingivalis*, *Treponema denticola*
Real Time Polymerase Chain Reaction (PCR)	Amplification of species-specific genetic sequences	High sensitivity and specificity; culture-independent	Requires laboratory infrastructure; risk of contamination	Quantitative monitoring of target pathogens; assessment of treatment efficacy
DNA–DNA hybridization (checkerboard method)	Hybridization between immobilized DNA probes and target DNA sequences	High-throughput analysis of multiple samples and species	Requires specialized equipment and expertise	Longitudinal epidemiological studies; microbiota shift analysis
LAMP (loop-mediated isothermal amplification)	Isothermal amplification of target genes	Simple, rapid, portable	Complex primer design; limited multiplexing	Point-of-care screening for specific pathogens
Microarray	Hybridization of sample DNA (16S ribosomal RNA targets) to hundreds–thousands of arrayed probes	High throughput (many species × many samples)	Fixed probe set—misses novel/strain-level variation; Requires good DNA quality; lab-based	Profiling established oral panels (health vs. periodontitis); monitoring post-therapy shifts
Next-Generation Sequencing (metagenomic analysis)	High-throughput sequencing of total microbial DNA	Broad-spectrum, culture-independent; detects rare and novel species	Expensive; requires bioinformatics expertise	Basic microbiome research; discovery of novel periodontal pathogens
FISH (fluorescence in situ hybridization)	Fluorescent probe hybridization to visualize target bacteria in situ	Provides spatial distribution information	Lower sensitivity compared with PCR	Visualization of plaque biofilm architecture

**Table 2 ijms-26-10172-t002:** Studies evaluating enzymatic activity in relation to probing pocket depth and disease status.

Reference	Sample Size	Enzymatic Activity Assessed	Relationship with PPD	Association with Disease Status
Lamster et al., 1988 (ref. [29])	36 patients	GCF lysosomal enzymes (β-glucuronidase, etc.)	Increased activity at sites with higher PPD	Whole-mouth β-glucuronidase distinguished progressive disease
Loesche et al., 1990 (ref. [30])	Not specified	BANA (subgingival plaque)	BANA-positive teeth showed greater PPD reduction after SRP + MTZ	Associated with spirochetal infection and responsive periodontitis
Grisi et al., 1998 (ref. [31])	28 patients, 513 sites	BANA (subgingival plaque)	Deeper pockets showed higher BANA positivity (100% at ≥8 mm)	Positive sites associated with periodontitis
Usui et al., 2021 (ref. [32])	347 adults	TLP (tongue swab)	Not site-level; disease-level only	High values associated with severe periodontitis (AUC 0.83, sens. 83%, spec. 77%)
Iwasaki et al., 2020 (ref. [33])	105 adults	TLP (tongue swab)	Not site-level; disease-level only	Excellent diagnostic ability for severe periodontitis (AUC 0.93)
Mikami et al., 2025 (ref. [36])	30 patients	TLP (subgingival plaque)	Strong correlation with PPD (ρ = 0.80, *p* < 0.001)	Also correlated with *P. gingivalis* counts

PPD, probing pocket depth; TLP, trypsin-like protease; GCF, gingival crevicular fluid.

**Table 3 ijms-26-10172-t003:** Studies evaluating Immunochromatographic devices for *P. gingivalis*: associations with PPD and disease status.

Reference	Sample Size	Relationship with PPD	Association with Disease Status
Imamura et al., 2015 (ref. [41])	63 chronic periodontitis + 28 healthy	Device score positively correlated with PPD (r = 0.317, *p* < 0.01)	Sensitivity 96.2%, specificity 91.8% vs. PCR; no positives among healthy controls at cut-off ≥0.25
O’Brien-Simpson et al., 2017 (ref. [42])	50 periodontitis patients + 50 healthy	Device result correlated with probing depth (r = 0.695, *p* < 0.01)	Positive test strongly associated with high salivary *P. gingivalis* (>10^5^ cells/mL); overall accuracy 94.0% (Se 95.0%, Sp 93.3%)
Yamanaka et al., 2024 (ref. [43])	72 chronic periodontitis + 53 healthy	Device score significantly higher at deeper pockets; positive correlation with PPD	Periodontitis classification: AUC 0.73; IC score correlated with qPCR counts (r = 0.73)

PPD, probing pocket depth; AUC, area of under curve; IC, immunochromatography.

**Table 4 ijms-26-10172-t004:** Quantitative PCR detection of periodontal pathogens in relation to probing depth and disease status.

Reference	Sample Size	Relationship with PPD	Association with Disease Status
Lyons et al., 2000 (ref. [53])	20 periodontitis patients, 20 healthy controls	Higher *P. gingivalis* counts by qPCR in deep pockets; qPCR more sensitive than culture	qPCR discriminated periodontitis patients from controls based on *P. gingivalis* load
Boutaga et al., 2003 (ref. [55])	59 periodontitis patients (181 subgingival samples)	qPCR more frequently detected *P. gingivalis* in deeper sites than culture	Higher detection in periodontitis compared with health
Lau et al., 2004 (ref. [56])	40 periodontitis patients	Bacterial loads by qPCR increased with pocket depth for *A. actinomycetemcomitans*, *P. gingivalis*, *T. forsythia*	qPCR distinguished disease from health more effectively than culture
Jervøe-Storm et al., 2005 (ref. [57])	33 periodontitis patients	qPCR detected higher levels of pathogens in deeper pockets compared with culture	Higher prevalence of red complex bacteria in periodontitis
Nonnenmacher et al., 2004 (ref. [58])	21 periodontitis patients, 20 healthy controls	qPCR showed higher pathogen counts in diseased sites with deeper PPD	Pathogen levels significantly associated with periodontitis vs. health
Kuboniwa et al., 2004 (ref. [59])	18 periodontitis patients, 18 healthy controls	TaqMan qPCR showed correlation between pathogen counts and probing depth	Strong association between bacterial load and periodontal status (disease vs. healthy)

PPD, probing pocket depth; qPCR, quantitative polymerase chain reaction.

## Data Availability

No new data were created or analyzed in this study. Data sharing is not applicable to this article.

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
