# Peer review of "Measuring the Invisible: Microbial Diagnostics for Periodontitis—A Narrative Review"

_ijms, 2025, doi:10.3390/ijms262010172_

Round 1

Reviewer 1 Report

Comments and Suggestions for Authors

The paper is full of information about tests used in diagnostic of periodontal diseases but some additional issues could improve the value of the paper.

Introduction - a medical interview is missing as as a key point in periodontitis diagnosis. This aspect should be more described.

There is a lack of justification of the purpose of the work.

In the presented methods, the aboratory blood test is missing - it is a mirror of inflamation.

Discussion - more clinical aspects are missing in relation to this all desribed tests.

And it is not clear what means " in our research" mentioned few times - this is not an original paper.

Comments on the Quality of English Language

I think, it is better to write your article to use the passive voice.

Author Response

We sincerely thank the reviewers for their thoughtful and constructive comments on our manuscript. We have carefully considered all the suggestions and revised the manuscript accordingly. Below, we provide a detailed, point-by-point response to each comment.

Reviewer 1

Comment 1

Introduction - a medical interview is missing as as a key point in periodontitis diagnosis. This aspect should be more described.

Response:
Thank you very much for your valuable comment. As you correctly pointed out, patient interview is indeed important in the diagnosis of periodontitis. However, the focus of this review article is not on the general “diagnosis of periodontitis,” but rather on “microbiological diagnostic approaches in periodontal tissues.” Therefore, we did not describe patient interview in detail in the Introduction. Nevertheless, we have now added patient interview, along with probing pocket depth (PPD) and radiographic examination, as part of the general diagnostic methods for periodontitis. (P2 L64)

Comment 2

There is a lack of justification of the purpose of the work.

Response:

Thank you for your comment. The purpose of this review is stated in the last paragraph of the Introduction. In addition, we have further elaborated on periodontal bacterial testing, particularly regarding its application in actual clinical treatment, to clarify the significance of such testing as addressed in this review. (P2 L89 – P3 L95)

Comment3

In the presented methods, the aboratory blood test is missing - it is a mirror of inflamation.

Rsponse:

Thank you very much for your thoughtful comment. In this review, we chose to focus specifically on periodontal bacterial testing. We fully agree that the evaluation of inflammation is important in the pathophysiology of periodontitis, and that blood tests, such as those measuring serum CRP levels, can provide valuable information. However, since these approaches are outside the scope of bacterial testing, we did not address them in the present review.

Comment 3

Discussion - more clinical aspects are missing in relation to this all desribed tests.

Response:
Thank you for your valuable comment. We have added a discussion of the clinical aspects of these tests in the Discussion section, specifically in section 3.10, “Practical Implications.” In this added section, we highlight how these diagnostic approaches may be applied in actual periodontal care. P14 L517 – L528

Comment4

And it is not clear what means " in our research" mentioned few times - this is not an original paper.

Response:

Thank you for your helpful comment. We have revised the text to avoid expressions such as “in our research group,” which may place unnecessary emphasis on the authors’ own work, and have instead rephrased it in more general methodological terms. P4 L120, L137-139, and P6 L177-181

Reviewer 2 Report

Comments and Suggestions for Authors

This review article reports that microbiological testing for periodontitis is most effective when it is quantitative/functional and standardized. By combining various tests with activity and host response readouts, we can obtain the best information for periodontal treatment.

I think this manuscript has provided us with useful information; however, a few points could be improved as follows.

  1. I recommend that the authors indicate the comparison data between each diagnostic method and actual clinical data, for example, PISA or alveolar bone absorbance.
  2. The authors should indicate each report sample of each diagnostic method, which means the authors should give us the information on how and what kinds of information/results we can get by these methods.

Author Response

We sincerely thank the reviewers for their thoughtful and constructive comments on our manuscript. We have carefully considered all the suggestions and revised the manuscript accordingly. Below, we provide a detailed, point-by-point response to each comment.

Reviewer 2

Comment 1 and 2
I recommend that the authors indicate the comparison data between each diagnostic method and actual clinical data, for example, PISA or alveolar bone absorbance.

The authors should indicate each report sample of each diagnostic method, which means the authors should give us the information on how and what kinds of information/results we can get by these methods.

Response:

Thank you very much for your helpful comments. Although we did not find studies that clearly addressed the relationship between the diagnostic methods discussed in this review and clinical parameters such as alveolar bone level or PISA, we have now organized a table that summarizes the associations with probing pocket depth (PPD) and disease status. To make this clearer, the table is structured by each diagnostic method, including enzyme activity assays, quantitative PCR, immunochromatography, and DNA checkerboard hybridization. We have also added information on sample sizes for reference. We believe that this addition makes the clinical relevance of these diagnostic methods clearer and enhances the usefulness of the review for readers. P5L164, P6L191 and P8 L243